# Feedback control of organ size precision is mediated by BMP2-regulated apoptosis in the *Drosophila* eye

Tomas Navarro[1], Antonella Iannini[1], Marta Neto[1¤], Alejandro Campoy-Lopez[1,2], Javier Muñoz-García[3], Paulo S. Pereira[4], Saúl Ares[5]*, Fernando Casares[1]

**1** CABD, CSIC/Universidad Pablo de Olavide, Seville, Spain, **2** ALMIA, CABD, CSIC/Universidad Pablo de Olavide, Seville, Spain, **3** Grupo Interdisciplinar de Sistemas Complejos (GISC) and Departamento de Matematicas, Universidad Carlos III de Madrid, Leganes, Spain, **4** I3S, Instituto de Investigação e Inovação em Saude, Universidade do Porto; IBMC- Instituto de Biologia Molecular e Celular, Universidade do Porto, Porto, Portugal, **5** Grupo Interdisciplinar de Sistemas Complejos (GISC) and Centro Nacional de Biotecnologia (CNB), CSIC, Madrid, Spain

¤ Current address: Instituto de Neurociencia, CSIC/Universidad Miguel Hernández, Sant Joan d'Alacant, Spain
* saul.ares@csic.es (SA); fcasfer@upo.es (FC)

**Data Availability Statement:** Quantitative raw data used for the analyses presented in the main and supplementary figures and that might be required to reanalyze the data reported in this paper are

## Abstract

Biological processes are intrinsically noisy, and yet, the result of development—like the species-specific size and shape of organs—is usually remarkably precise. This precision suggests the existence of mechanisms of feedback control that ensure that deviations from a target size are minimized. Still, we have very limited understanding of how these mechanisms operate. Here, we investigate the problem of organ size precision using the *Drosophila* eye. The size of the adult eye depends on the rates at which eye progenitor cells grow and differentiate. We first find that the progenitor net growth rate results from the balance between their proliferation and apoptosis, with this latter contributing to determining both final eye size and its variability. In turn, apoptosis of progenitor cells is hampered by Dpp, a BMP2/4 signaling molecule transiently produced by early differentiating retinal cells. Our genetic and computational experiments show how the status of retinal differentiation is communicated to progenitors through the differentiation-dependent production of Dpp, which, by adjusting the rate of apoptosis, exerts a feedback control over the net growth of progenitors to reduce final eye size variability.

## Introduction

The size of some organs is remarkably precise [1]. Although this precision is a functional need, it is not easily explained [2,3]. Most organs are specified as small primordia whose growth is exponential. However, exponential growth is prone to amplify the noise that is inescapable to biological processes. This fact has led to the proposal that organ development must be subject to mechanisms of feedback control [3], which, in the engineering sense, reduce their internal "developmental instability" [4] and increase the precision with which organs reach their

available as supplementary data. The macros for the quantification of apoptosis/proliferation signal from confocal images is provided as supplementary material.

**Funding:** PID2019-109320GB-100 to SA and JM-G; PID2022-142185NB-C21 to SA; PID2022-142185NB-C22 to JM-G; PGC2018-093704-B-I00 and PID2021-122671NB-I00 to FC. Agencia Estatal de Investigacion, SPAIN: https://www.aei.gob.es/ The CABD and CNB also received funding from the Spanish Agencia Estatal de Investigacion through the "Maria de Maeztu" and "Severo Ochoa" Centres of Excellence Program through grants CEX2020-001088-M and SEV 2017-0712, respectively, from Agencia Estatal de Investigación, SPAIN: https://www.aei.gob.es/. We acknowledge support from the Scientific Network LifeHub. CSIC funded by the Consejo Superior de Investigaciones Científicas (CSIC), SPAIN: https://www.csic.es/en. FCT (Fundaçao para a Ciencia e a Tecnologia, Portugal). Grant number: EXPL/BIA-BID/0267/2021) to PSP. The funders had no role in study design, data collection and analysis, decision to publish, or preparation of the manuscript.

**Competing interests:** The authors have declared that no competing interests exist.

**Abbreviations:** dad, *daughters against-dpp*; Dpp, *decapentaplegic*; FAi, fluctuating asymmetry index; Hh, *hedgehog*; Hth, homothorax; ROI, region of interest; sFAi, *signed* FAi; tkv, *thickveins*.

species-specific size. Some organs that undergo constant tissue renewal, like the mammalian olfactory epithelium, maintain their size homeostasis through integral feedback [5]. Some other organs, though, grow to a size that, once reached, cannot be further modulated. This is the case of insects. After their last molt, the adult organs cease growth and then become covered in a rigid chitin exoskeleton that precludes further size regulation. The eye of *Drosophila* fruit flies serves as a good example of this type of organ growth: In *Drosophila*, the end point of eye development is reached when all retinal progenitor cells differentiate, as differentiation is accompanied by the exit of the cell cycle. Three features should have resulted in a strong evolutionary pressure to maximize the precision in eye size: First, size impacts vision directly, as image resolution and contrast sensitivity is proportional to the number of light sensing units in the eye [6]; second, making and maintaining the eyes is energetically very expensive [7], so there is a pressure to match eye size to vision needs; and third, left and right eyes must survey a symmetrical part of the space, so eye asymmetry, which could be driven by developmental noise, should be minimized. In this paper, we investigate the mechanism by which the *Drosophila* eye ensures its size precision as an example of organ size control.

The development of the *Drosophila* eye takes place as a wave of differentiation crosses the eye primordium during the last larval stage (Fig 1A and 1B). As the wave moves, proliferative progenitors anterior to the wave are recruited as differentiating retinal cells that exit the cell cycle. Final eye size is attained when the differentiation wave reaches the anterior-most region of the primordium: With no remaining progenitors, which are the source of growth, the process ends. The movement of the wave is driven primarily by 2 secreted signaling molecules. The first of them is Hh (*hedgehog*). Hh's major role is the induction of differentiation of progenitor ("G") cell into retinal ("R") cells [8], a role that is shared by its homologue Shh in the differentiation of the ganglion cells of the vertebrate retina [9]. Hh, initially expressed adjacent to the eye primordium, contributes to initiate R differentiation. Once initiated, R cells produce Hh themselves [10,11], resulting in the self-sustained progression of a differentiation wave [12]. The second signal is a BMP2/4 homologue, Dpp (*decapentaplegic*). *dpp* expression is induced by Hh in the newly recruited retinal cells ("Rn") [13,14]. Dpp, by repressing the progenitor transcription factor Hth (*homothorax*), increases the propensity of progenitors to enter differentiation [15,16]. However, and in contrast with Hh, whose expression is maintained in differentiated R cells, Dpp is expressed only transiently: As newly recruited Rn cells progress along their differentiation to R, *dpp* expression is turned off [13,17,18]. Thus, a stripe of Dpp marks the front of the differentiation wave (Fig 1A). This process is schematically represented in Fig 1C.

## Results

To ensure that the target, species-specific final eye size is reached with precision, the rates of growth and of differentiation need to be coordinated within the eye primordium [6,19,20]. For example, a transient halt of the differentiation wave would produce some excess of progenitor cells and a concomitant deviation from the target size—unless this halt were communicated to the progenitors as a negative feedback on their rate of growth. This feedback could be implemented either by slowing down progenitor proliferation or by curving the excess of progenitor cells through cell death (we do not consider a change in cell size as this change would alter the size of the ommatidia and, thus, their optical properties). We noticed that, in wild-type control eye primordia, the apoptosis signal detected using an antibody against the activated form of Caspase-3 is usually very low. However, if an antibody against the cleaved, active form of the terminal caspase Dcp-1 [21] is used instead, we detected a stripe of apoptotic cells just anterior to the differentiation wave-front (S1 Fig). Likely, the activation of Cas-3 is more brief than the

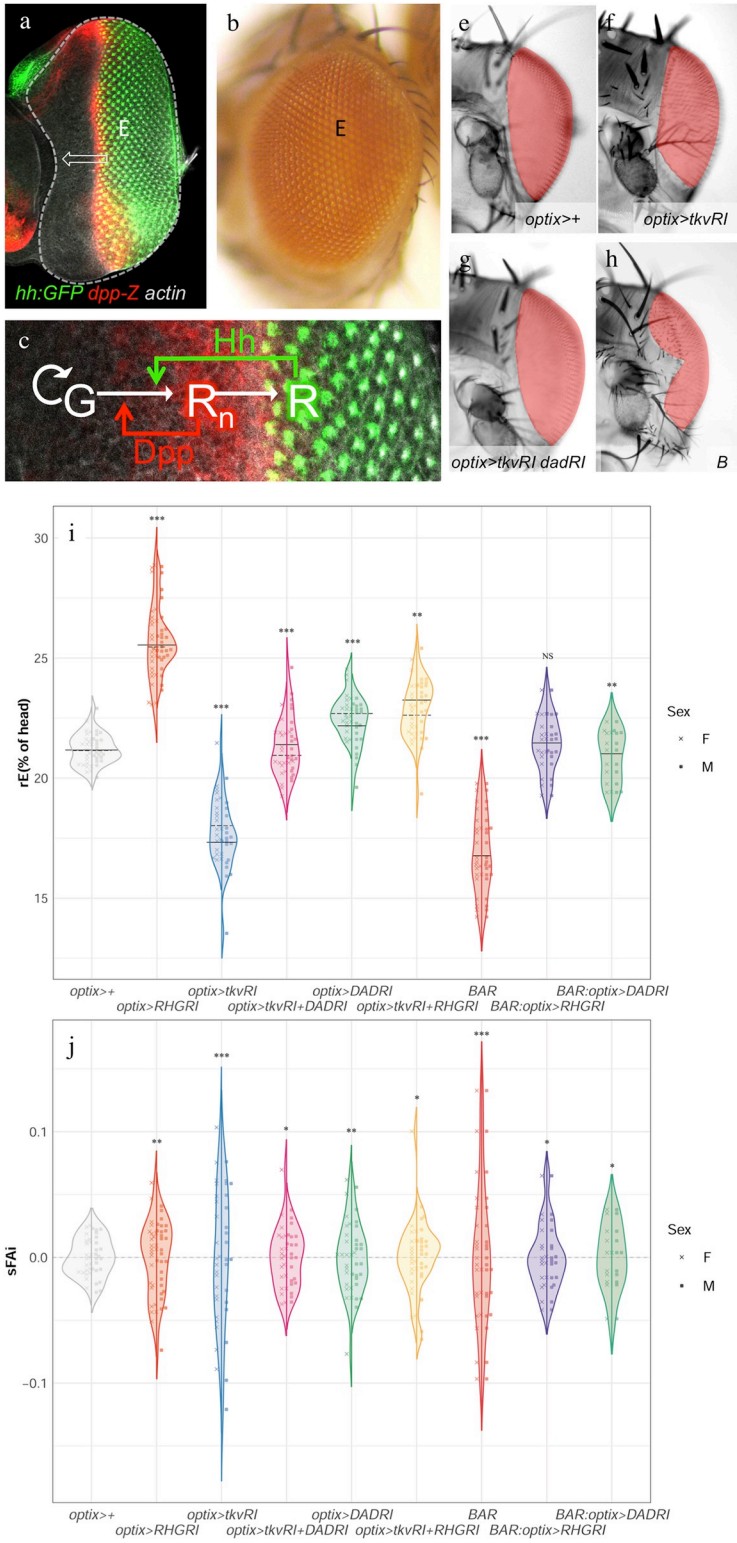

**Fig 1. Eye development and the effects of different genetic manipulations on the distribution of eye size ("rE")
and signed Fluctuating Asymmetry index ("sFAi").** The eye primordium ((**a**) "E", outlined) develops into the adult
eye (**b**) through the action of a differentiation wave that sweeps across it (arrow). (**a**) Eye primordium ("E") of a *Hh*:
*GFP*; *dpp-Z* larva. The differentiation wave-front expresses *dpp-Z*. Posterior to it further differentiated photoreceptors
lose *dpp* expression and express *hh*. (**b**) Lateral view of an adult *Drosophila* eye. (**c**) The recruitment of proliferative

progenitor cells ("G") into early differentiating (*dpp*-expressing) retinal cells ("Rn") and then into terminally differentiating (non-*dpp*-expressing) retinal cells ("R") is controlled by 2 signaling molecules, Hh and Dpp. (**e**-**h**) Frontal views of female heads of the indicated genotypes, with the eyes colored for better visualization. (**e**) Control, *optix>+*, (**f**) *optix>tkvRNAi*, (**g**) *optix>tkvRNAi + dadRNAi*, (**h**) *Bar (B)* mutant. See main text for details. (**i, j**) rE (**i**) and sFAi (**j**) distributions of the indicated genotypes. Each violin plot includes the male (left) and female (right) distributions, which were not significantly different from each other in any genotype. The median of each distribution is indicated (solid and dashed line: male and female, respectively). The predicted rE median for each genotype and the statistical model from which they are derived can be found in S1 Table. The variances of each genotype were compared against the variance in the control group (*optix>+*) in rE and sFAi (S1 Table). (\*): $p < 0.05$; (\*\*): $p > 0.01$; (\*\*\*): $p < 0.001$. Eye area is measured relative to the total head area and expressed as a percentage in this and subsequent figures. The data underlying the graphs shown in the figure can be found in "Fig 1_S6Fig 1_data" in the Supporting information file S1 Raw Data.

activation of Dcp-1 during the apoptotic process, making detection of apoptosis easier with Dcp-1 as a marker.

In any case, we were intrigued by this localized apoptosis happening in normal eye primordia and asked what its role could be. To answer this, we blocked apoptosis by driving, specifically anterior to the differentiation wave-front, a polycistronic RNAi against *reaper*, *hid*, and *grim*, the 3 major proapoptotic genes in *Drosophila* (*optix>Reaper,Hid,Grim-RNAi*, abbreviated *optix>RHGRI* [22]). These flies showed 2 major phenotypic changes: First, the median eye size ("rE" = average intraindividual eye area relative to head area, expressed as percentage) was larger than those of controls (*optix>+*), and, second, the distribution of eye sizes was also significantly broader (Fig 1I and S1 Table). We do not detect significant changes in ommatidial size in these (and other; see below) genotypes, indicating that the alterations in eye size are attributable to variation in ommatidial number (S2 Fig). When we measured the apoptotic index of *optix>RHGRI*, it was smaller than that of *optix>+* and very close to zero (Fig 2; the apoptotic index is measured as the ratio of Dcp-1 positive pixels relative to the total pixels of the region anterior to the morphogenetic furrow of the eye primordium; see Materials and methods). The increased dispersion in the distribution of eye size is indicative of a loss in size precision. The fluctuating asymmetry index (FAi), which measures the disparity between left and right organs within the same individual, has also been frequently used as a measure of precision as it captures the sensitivity of the process to developmental noise [23]. The FAi is normally calculated as the absolute value of the normalized difference between the size of the left (L) and right (R) organs from the same individual. Here, we use the *signed* FAi ("sFAi") to have into account the sign of the asymmetry, $(L−R)/(L+R)$, which allows to compare the asymmetry across different genotypes based on the amplitude of the distribution of this metric for each of them (Fig 1J and S1 Table). In apoptosis-blocked individuals, the variance of sFAi increased significantly as well (Fig 1J and S1 Table). To control for potential nonspecific effects of RNAi overexpression on eye size and its variability, we drove a "neutral" RNAi against the fluorescent protein Cherry (*optix>CherryRI*) and observed that it did not affect neither of the 2 parameters when compared to control (*optix>+*) or wild-type (*Oregon-R*) flies (see S3 Fig).

These results pointed to a role of developmental apoptosis in controlling eye size and its precision. But, in order to be involved in such control, the apoptotic rate in progenitors should be coupled to the rate of eye differentiation. This communication between progenitors and differentiating cells might be exerted by either of the 2 major signals in eye development: Hh and Dpp.

Intuitively, if one of these 2 signals were to be used to convey feedback information to progenitors about the status of retinal differentiation, Dpp would seem a better candidate. Although both Hh and Dpp regulate progenitor gene expression [24], Dpp, by being expressed transiently, can be better used as readout of the *rate* of retinal differentiation. In contrast, Hh is constantly produced in R cells, regardless of whether the wave is moving or not. If Dpp were used in a feedback control mechanism, it would be expected that the attenuation of its

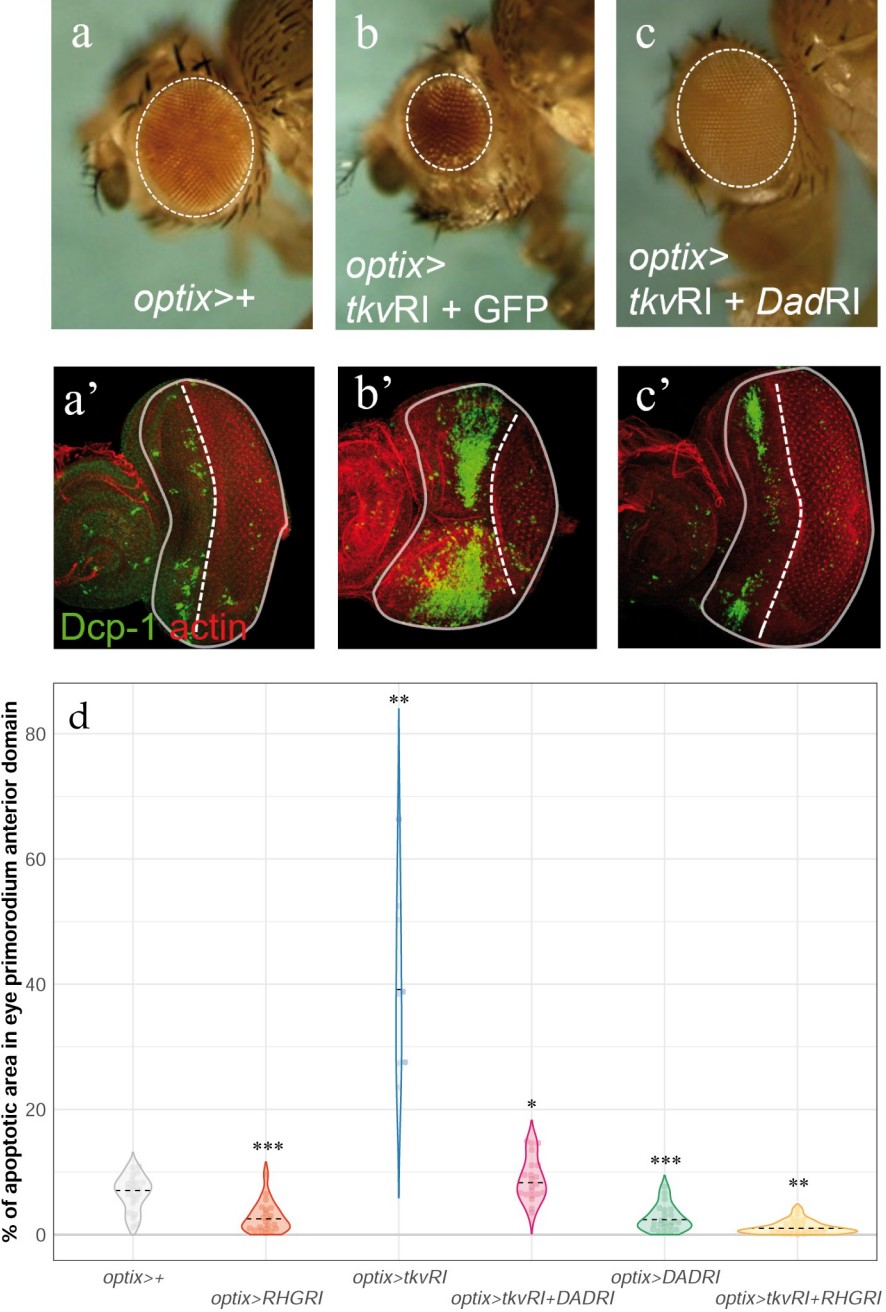

**Fig 2. Eye size reduction and apoptosis induced by attenuation of the Dpp signaling pathway can be reverted by blocking Dad.** Lateral views of adult eyes (**a-c**) and L3 eye discs, stained with Dcp-1 (and counterstained with rhodamine phalloidin, "actin") (**a'-c'**) of the indicated genotypes. (**d**) Distribution of apoptosis intensity in L3 discs of the indicated genotypes. Blocking apoptosis driving the UAS-RHG-RNAi reduces the apoptosis caused by *tkv* attenuation (*optix>tkvRI + RHGRI*). The statistical analysis corresponding to these data can be found in S2 Table. (*): $p < 0.05$; (**): $p < 0.01$; (***): $p < 0.001$. The data underlying the graphs shown in the figure can be found in "Fig 2_data" in the Supporting information file S1 Raw Data.

signaling pathway in progenitors should lead to a decrease in final eye size precision. In addition, prior work on *dpp* mutant alleles had shown an increase in apoptosis in the mutant eye primordia [17,25], reinforcing the idea that Dpp could be this coupling signal.

To test this point, we attenuated the Dpp pathway specifically in eye progenitors during development, by targeting the expression of an RNAi against the Dpp receptor *tkv* (*thickveins*) [26] in these cells (*optix>tkvRI*). This treatment caused 2 effects: First, the median eye size was reduced (Fig 1E, 1F and 1I and S1 Table). This effect—eye reduction—had been described previously for mutant alleles of *dpp* [27] and explained as a consequence of Dpp's requirement in progenitor cell recruitment [24,28]. The eye size reduction was concomitant to a dramatic increase of both the dispersion of the eye size distribution and of its sFAi (Fig 1J and S1 Table), something expected if Dpp were involved in feedback control. To confirm that this phenotype resulted from the attenuation of the Dpp pathway, we simultaneously drove expression of RNAis against *tkv* and against *dad* (*daughters against-dpp*), a Smad molecule that acts as a feedback inhibitor of the pathway [29,30]. In individuals of this genotype (*optix>tkvRI+-DadRI*), the eyes almost recovered their control size (Fig 1E, 1F and 1I and S1 Table). Similarly, the dispersion of eye size distribution as well as the sFAi decreased towards values closer to control, although not fully (Fig 1J and S1 Table). The attenuation of the *tkv-RNAi* effect cannot be attributed to its weaker expression caused by the introduction of a second UAS sequence (*UAS-Dad-RNAi*) in the genotype (see S4 Fig).

Next, we confirmed that attenuation of the Dpp pathway (*optix>tkvRI*) caused a dramatic quantitative increase in Dcp-1 apoptotic signal (Figs 2A, 2B, and S8 and S2 Table), while the mitotic rate of progenitors remained unaffected (S7 Fig), and that this increase in apoptosis was rescued by coexpression of the Dad-RNAi (*optix>tkvRI+DadRI*) (Fig 2C). The expression of *UAS-RHGRI* alone or in combination resulted in the reduction of apoptosis almost completely (Fig 2D).

Therefore, we expected that any perturbation affecting the coupling between retina and progenitor apoptosis would not be able to fully rescue the size precision observed in control flies (both the dispersion of the eye size distribution or the sFAi). This was the case in all the experiments in which the feedback between retina-produced Dpp and apoptosis was disrupted, either by artificially up-regulating the Dpp pathway (expressing *dad-RNAi*) or by blocking apoptosis (expressing *RHG-RNAi*) (Fig 1I and 1J and S1 Table).

To further substantiate our results, we decided to analyze the phenomenon of eye size control in another mutant situation in which the *dpp* pathway was affected. We chose the *Bar* (*"bar-eyed"*) mutant. *Bar*-mutant eyes are reduced [31] (Fig 1E and 1H), and this size reduction is associated with the control of *dpp* expression and an increased apoptosis during larval development [32]. In *Bar* eye primordia, we detected a fragmented *dpp* expression, monitored by the *dpp-Z* transcriptional reporter, and an increased Dcp-1-positive apoptosis (S5 Fig). *Bar* eyes are smaller and more variable (Fig 1I) as well as more asymmetric (Fig 1J) than control flies (see also S1 Table). Similarly to what happened in our previous experiments, in which the Dpp pathway was attenuated directly by targeting the Dpp receptor with an RNAi, either blocking apoptosis (*Bar; optix>RHGRI*) or derepressing the Dpp pathway (*Bar; optix>DadRI*) partly rescued eye size (Fig 1I) and reduced both eye dispersion (Fig 1I) and their fluctuating asymmetry (Fig 1J and S1 Table).

The results obtained so far indicated that a major role of Dpp during eye development was the regulation of developmental apoptosis. Losing this regulation, by either blocking apoptosis directly or by "disconnecting" retina and progenitors by the attenuation of Dpp signal (by *tkvRI* expression or in *Bar* mutants), led to a decreased size precision. Therefore, not only the size of the eye but also its precision is under the control of the Dpp pathway.

The fact that the effects on eye size and precision caused by *dpp* signaling attenuation were almost fully rescued by preventing apoptosis pointed to the regulation of this process as a key phenomenon. In order to explore further how the regulation of apoptosis impinges on eye size, we built and analyzed a mathematical model of the process. With this model, we first

explored whether our description of the biological phenomenon is complete by testing if the model recapitulates the biological results when parameterized using experimental measurements.

Our model comprises G (progenitors), Rn (newly differentiated retinal cells, expressing *dpp*), and R (differentiating retinal cells, no longer expressing *dpp* but producing Hh) cells as variables, as well as their transformations: proliferation of progenitors ("self-renewal": G → G + G), apoptosis of progenitors (G → ∅), initial differentiation of progenitors into *dpp*-expressing early retinal cells (G → Rn) and final differentiation of these cells into *hh*-expressing photoreceptors (Rn → R) (Fig 3A and 3B), and includes noise in all 4 differentiation steps. Its full

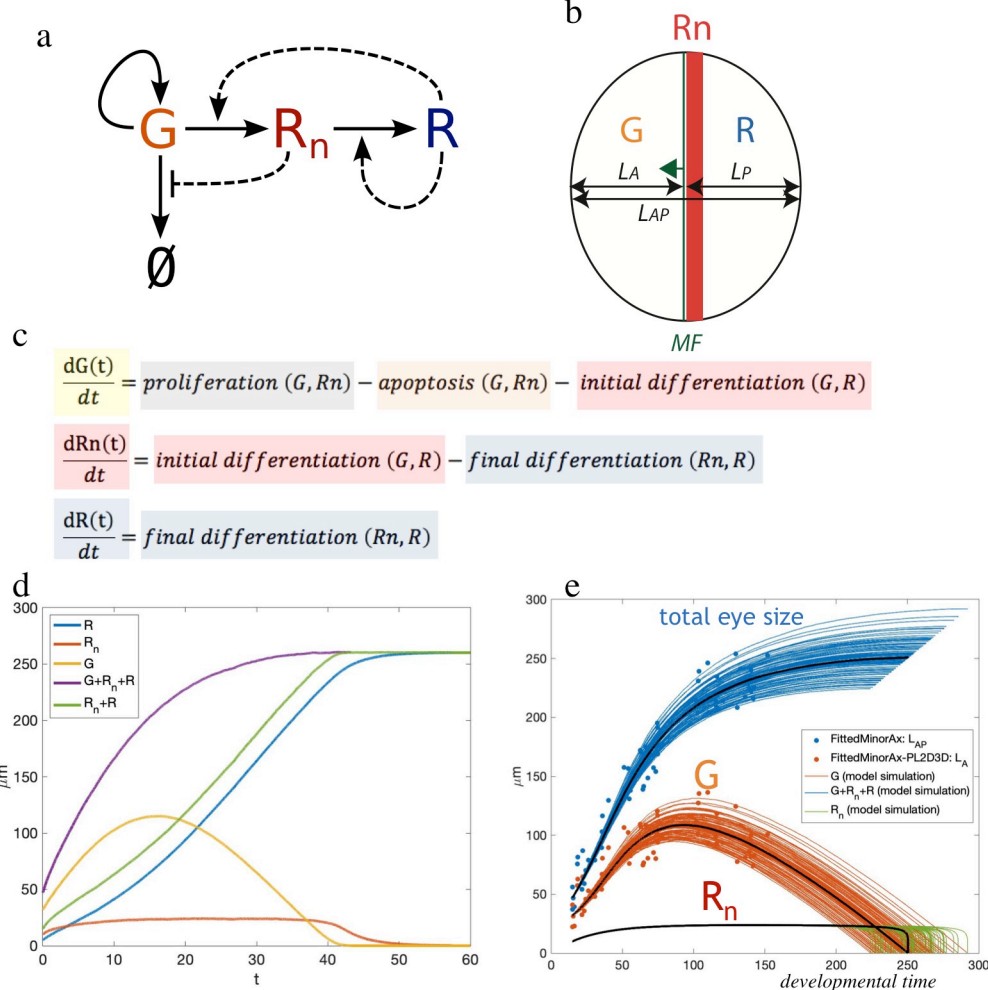

**Fig 3. Overview of the model of the GRnR including apoptosis (∅). (a)** Schematics of the process. R cells stimulate the differentiation process (by expressing Hh). Rn cells negatively regulate apoptosis (by producing Dpp). G cells can divide, die, or progress further into differentiation by transiting to Rn. **(b)** This process, schematized in space. MF: morphogenetic furrow is the differentiation wave-front that moves (arrow) from posterior (right) to anterior (left). $L_A$, $L_P$ are the maximum distances anterior and posterior to the MF, respectively. The $L_P$ can be used as a proxy for developmental time: As the MF moves with constant speed, $L_P$ increases as time passes. $L_A$ corresponds to G cells. $L_P$ comprises Rn (orange bar) + R cells. **(c)** "Qualitative" model. **(d)** Typical temporal profiles of all populations as predicted by the model. **(e)** Fit of the model to experimental data ($L_{AP}$ as minor axis of elliptical fits of eye primordia (blue points) and $L_A = L_{AP} - L_P$ (orange points)). Each curve is an independent realization produced with the *optix>+* parameter set (see main text and S1 Materials for details), differences between curves are an effect of the stochastic nature of the model.

description is available as S1 Materials. Conceptually, our model expresses the rates at which the G, Rn, and R cells change over time as in Fig 3c, where the variables influencing each of the steps (e.g., "proliferation") are within the parentheses (e.g., "G, Rn").

The key processes are modeled as follows: Apoptosis is a decreasing function of the Dpp-producing Rn cells, because the more Dpp, the less apoptosis, and is proportional to the number of progenitors (G). All G cells not entering apoptosis will be proliferating. As for the rate of G to Rn differentiation ("early differentiation"), we have considered 2 factors. First, the activation of differentiation would depend on an increasing function of the Hh-producing R cells. Second, the induction of G cells to differentiate is limited to the range of action of Dpp produced by Rn. Therefore, since beyond a certain point having more G cells will not result in increased differentiation (that is, those G cells out of the range of action of the signals will not be recruited), we modeled the function describing G growth as a saturating sigmoidal function, with the condition that once the G population has been exhausted, no further differentiation can occur. In addition to Dpp (represented by the action of Rn) also Hh has a contribution to the differentiation of G cells [16,24,33], which is represented by the action of the Hh-producing R cells. Finally, the final differentiation, or transition from Rn into R, can be expressed as being proportional to Rn, indicating that *Rn* differentiates into R cells with a constant pace, simplifying a complex process of successive cell induction leading to the formation of the ommatidial cell types [34]. However, eye size does not depend on this latter process, as the number of ommatidia and, therefore, the size of the eye, is directly related to the number of G cells that are recruited as Rn cells. The expressions of the relevant functions ($a(Rn)$, $f(G)$ and $h(R)$) can be found in the full description of the model in S1 Mathematical Model. These functions allow us to model the RNAi treatments as well.

With this model, we can calculate the evolution of all these cell populations over time (Fig 3D). First, we fitted the model to an experimentally determined growth curve of the eye primordium (Fig 3E; data from [35]). Since the differentiation velocity is about constant during most of development [35,36], the width of the R + Rn domain (the distance between the posterior margin of the primordium to the differentiation wave-front, called "posterior length", or "Lp") is a proxy for elapsed developmental time, and we used it as a time axis (as in [35]). The fitting demanded that the growth rate of G cells declined over developmental time, something that had been shown experimentally [36,37]. With these parameters, the fitting of the model to the data was very good, indicating that the model captured the biology underneath the control of eye size and its variability (Figs 3E, 4A, and 4B).

Despite the excellent qualitative agreement of theory and experiment, it is important to remark that this is a very minimalistic model. The concentrations of signaling molecules Dpp and Hh are not explicitly modeled; instead, we model their effect as being proportional to the width of the regions producing them, Rn and R, respectively. In doing so, we are formulating a *mean field* model: We do not model the spatial dynamics of morphogen gradients. Our rational for this simplification is not simplicity, but data availability: We have not measured those gradients, so their inclusion in the model would be unconstrained. In principle, this simplification would undermine the power of our model to reproduce experiments, but we found that this worked fine except for one aspect: Progenitor cells can only differentiate when they are close to the differentiation wave-front. In a *mean field* setting without an explicit modeling of space and gradients, there is no way to model this constraint from first principles. To restrict progenitor differentiation G  Rn to the vicinity of the morphogenetic furrow, we introduce an ad hoc length scale of differentiation modeled through a sigmoidal function with a single parameter, $K_G$. This allowed to capture a spatial effect in a phenomenological way, and as our sensitivity analysis of the model shows (see the "S1 Mathematical Model" supporting information), the model results are very insensitive to the exact value chosen for this parameter.

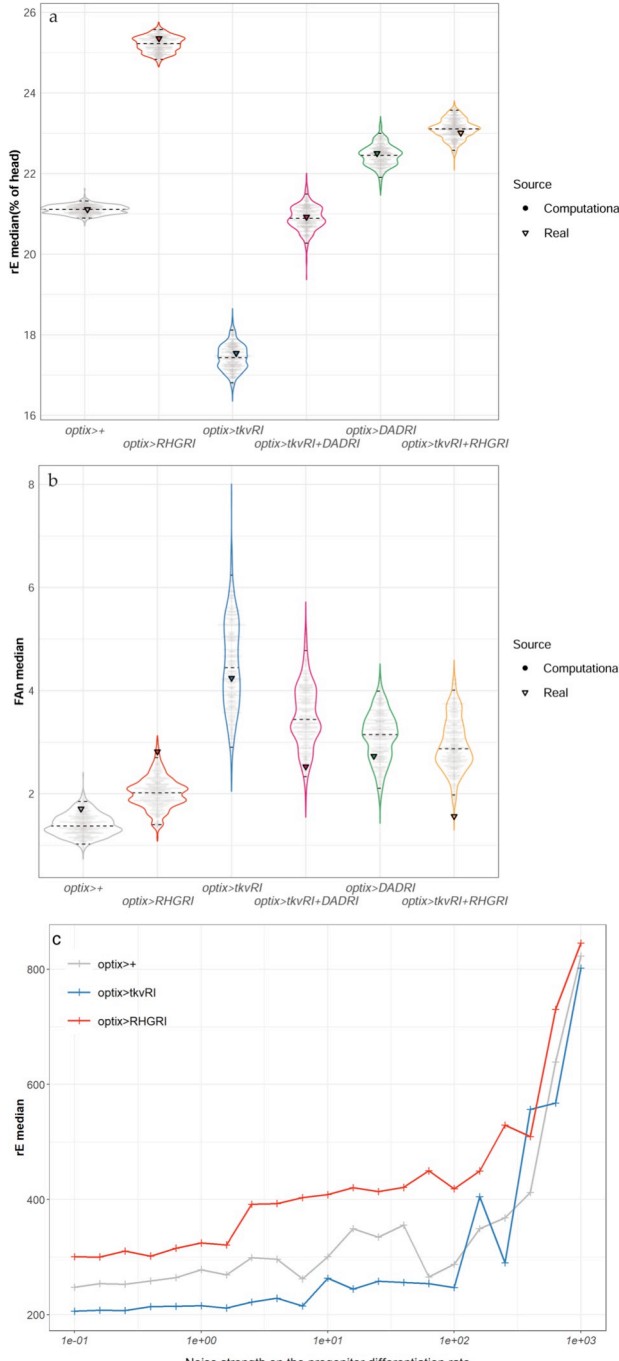

**Fig 4. Assessment of the degree of fit between experimental and computational results and effects of increasing noise strength on median eye size (rE).** Using our computational model, sets of 1,000 eye sizes were simulated for each condition and from these, samples of equal size to the experimental samples were taken to calculate the medians of rE (**a**) and interindividual FA (FAn) (**b**). The sampling was repeated 1,000 times to generate the distribution of medians shown in the graphs, where each point in gray corresponds to the median of one of those 1,000 samples ("Computational data"). The triangles represent the medians of the experimental results ("Real data") of all genotypes. The computational data were scaled such that the median rE of experimental control coincides with the center of the distribution for the computational control group. The experimental medians for both metrics fall within the respective computational distributions for all genotypes. This is especially striking for the rE medians, for which the experimental medians coincide almost perfectly with the most probable value of the computational median distributions. This fact indicates that the model captures the process of eye development in those different genotypes very accurately (See S1 Statistical Methods for further details). (**c**) If apoptosis is blocked in the computational model (by strongly reducing the

apoptotic rate of progenitors, equivalent to *optix>RHGRI*), eye size increases more rapidly than in the other 2 genotypes, in which apoptosis is allowed, until noise is too high (beyond 102), point at which allowing apoptosis or not becomes irrelevant. Computational results. Data for each point is from 200 simulations *per* genotype. The horizontal axis represents $1/V_p$ (see S1 Mathematical Model supporting information file). The data used in the graphs shown in the figure can be found in "Fig 4A,4B_data" in the Supporting information file S1 Raw Data.

Finally, since we do not model individual cells, our model does not capture the effects of cellular variability. However, our formulation of fluctuations using Langevin dynamics can be rigorously derived for the network of interactions shown in Fig 3A, and this framework is flexible enough to allow distinct modeling of the different sources of noise in our model. Table 1 points out some of the biological aspects in our model and the assumptions and simplifications we have made to formalize them mathematically.

Since the model was able to recapitulate well the dynamics of eye differentiation and the final eye size and its precision, we asked which conditions allow the system to reach robustly a final size. This can be done by carrying out a linear stability analysis of the system. The finalization of the process is attained when G = 0 and Rn = 0: Without any G and Rn cells left, the eye has reached its final size. The stability of this size means that for low G and Rn values, close to the end of the process, it is irreversible and effectively G and Rn will go to zero. Our analysis (see Linear stability analysis of the deterministic model in S1 Mathematical Model) shows that Dpp regulates stability through apoptosis, and Hh through differentiation: high apoptotic probability or differentiation rate favor stability. Actually, we derive a formula predicting that the maximum potential probability of apoptosis (that would be observed only in the absence of Dpp) of 50% would be enough to ensure stability by itself. This is not far from 40% of apoptosis determined experimentally in *optix>tkvRI*. Therefore, high propensity of apoptosis in progenitor G cells seems to be a requirement for robust size. Next, we used our model to predict the effects that noise would have on eye size in different genotypes with different rates of apoptosis.

We reasoned at the onset that regulating organ size would rely on the differentiating retina feeding information about its status on progenitor growth rates (something we have shown to occur through the modulation of apoptosis). Therefore, we simulated the effects of increasing

**Table 1. Simplifications in the formalization of some biological processes and how these have been modeled.**

| Biology | Simplification | Model |
|---|---|---|
| The domains of G, Rn, and R cells have a width along the anterior posterior axis of the eye primordium. | The size of the G, Rn, and R domains do not need to be referenced to a spatial axis as their relationships have a G>Rn>R polarity. | Three single-valued variables model the total width of the G, Rn, and R domains. |
| Cell state transitions induced by Rn and R cells are mediated by the signaling molecules Hh and Dpp, respectively. | Signaling intensity from Rn cells on G cells and from R cells on Rn and G cells can be represented without explicitly including Hh and Dpp as variables. | The intensity of the signaling is made proportional to the width of Rn and R (the producers of Dpp and Hh, respectively). |
| Hh and Dpp are distributed as decaying gradients in receiving cells such that cells respond to a signaling threshold. In this manner, signaling is restricted in space to a specific signaling distance or range. | There is no need to model the gradient explicitly, since it is the range of action the parameter that matters. | The model introduces a length scale to limit signaling to just a proportion of the G domain, which would correspond to those G cells closest to the signaling Rn and R cells. |
| Hh activates differentiation and Dpp inhibits apoptosis. | Use of sigmoidal functions including certain degree of cooperativity. | Activation and inhibitory Hill functions with exponent 2. |
| The process occurs across a field of cells, with individual cell variability. | The effect of fluctuations of the discrete cells can be added up and described by the fluctuations in the width of the regions they form. | Fluctuations/variability are captured through Langevin dynamics of the 3 variables G, Rn and R. |
| RNAi treatments alter phenotypes. | The effect of the RNAi treatments can be captured altering apoptosis dynamics. | Model RNAi treatments varying the apoptosis probability or the inhibitory strength of Rn. |

the noise in the recruitment of Rn cells ("initial differentiation"), which are the source of Dpp, on eye size and its variation in 3 genotypes: two in which apoptosis is possible (control (*optix>+*) and *dpp* signaling attenuated (*optix> tkv-RNAi*)) and one in which apoptosis is blocked (*optix>RHG-RNAi*) (Fig 4C). While increasing noise in Rn affects very little the median eye size in the 2 genotypes where apoptosis is allowed, it results in a steady increase of eye size if apoptosis is blocked (until the noise levels are very high, when allowing apoptosis or not becomes irrelevant). This computational result reinforces the idea that the *dpp*-sensitive apoptosis of progenitors is instrumental in maintaining the developmental stability of eye size.

In his paper of 1942, Waddington noted that while the morphological phenotype of wild-type strains was usually developmentally stable, or "canalized," "there is scarcely a mutant which is comparable in constancy with the wild type" [4]. That is, the phenotype of mutants is often "decanalized." We would expect decanalization of the phenotype if the mutants were affecting control feedback mechanisms. The observation that this decanalization is frequently exhibited by mutants with overt morphological alterations might indicate that most genes with major roles in the development of an organ are also part of the feedback control mechanism themselves or affect this mechanism. In the case of eye development, these would be mutants uncoupling retinal differentiation and progenitor growth, blocking Dpp reception or its control over apoptosis in progenitors. However, perturbations that maintain the feedback mechanism intact might result in phenotypic changes, but without major effects on precision. In the eye, one such perturbation could be the variation of progenitor proliferation rate. This is an input to the system and subject to the feedback control, but not itself part of the feedback mechanism. Indeed, the sensitivity analysis of our computational model indicates that, while eye size is highly sensitive to variations in progenitor growth rate, the FAi is much less so (see Fig 12 in S1 Mathematical Model)—that is, despite a major phenotypic change (eye size), the result of this perturbation should remain canalized. This perturbation of eye development can be induced in vivo by the genetic overexpression of the mitogenic cytokine and ligand of the JAK/STAT pathway *unpaired 1* (Upd) [38]. Expression of Upd using the eye-specific enhancer GMR results in overgrown eyes [38], about 1.4 times those of wild-type flies (S9A and S9B Fig). Although there is interindividual variability in rE (S9C Fig), the developmental noise of eye size, measured by its FAi, remains similar to that of control animals (S9D Fig). This latter result—the maintenance of organ size precision—is expected as long as apoptosis is operating under feedback control. Indeed, GMR>Upd discs exhibit apoptotic signal of progenitor cells (S9E and S9F Fig). Therefore, the mechanism controlling eye development can accommodate changes in final organ size while maintaining its precision as variations in progenitor cell proliferation do not interfere with the feedback control.

## Discussion

For organs to attain a final size with low variability, mechanisms of control must exist to reduce the effects of developmental noise. In this paper, we describe one such mechanism, in which the processes of eye growth and differentiation are coupled through a feedback control. In this feedback, the discrepancies between the rates of growth and differentiation of progenitors are adjusted through the modulation of the progenitor cells' propensity to enter apoptosis, so that any production of progenitors in excess is curved, as a proportion of these progenitors will not receive sufficient Dpp signaling to overcome apoptosis. In this way, it is the Dpp production, which, acting as a surrogate of the process of differentiation, feeds back onto the growth rate of progenitors, a rate that depends on their combined proliferation and apoptosis (this latter Dpp-signaling dependent). When analyzing our data, comparing the actual FAi versus the fluctuating asymmetry of randomized pairs generated from the same experimental

data (FAn), we observed that FAn was slightly larger than FAi, and this difference (FAn>FAi) was invariant across genotypes. This indicated a small but consistent effect of the individual. Therefore, although most variation in size can be attributed to organ-autonomous variation, the 2 eyes from an individual are slightly correlated (S6 Fig). This is expected though, as it has been shown that, besides organ-autonomous mechanisms of growth control, the growth discrepancies among organs in an individual affect the systemic growth of the organism through neuroendocrine communication mediated by insulin-like peptides [39,40]. In addition, each larva might be exposed to small microenvironmental variations during its growth [41]. Still, our results indicate that the feedback mechanism we have described operates mostly in an organ-autonomous manner.

Our experiments show that a major role of *dpp* signaling during eye development is the regulation of apoptosis, since the effects on eye size caused by attenuating this pathway are almost fully recovered just by blocking it. This role is very different from the function in cell differentiation that has normally been attributed to *dpp* during eye development [28]. We note, however, that in our experiments, by using RNAi, we are attenuating, rather than fully abrogating, *dpp* signaling. This could mean that the low levels of signaling presumably remaining could still participate in cell fate specification. This would be similar to the situation found in the wing disc, in which different signaling thresholds are important for distinct functions, with high *dpp* signaling being necessary for wing patterning, while lower levels sufficing for tissue growth [42–44]. Recently, the Kumar laboratory showed that the main role of the Pax6 gene *eyeless (ey)*, a gene essential for normal eye development [45], is the activation of *dpp* transcription [46]. Interestingly, loss of *ey* function causes abundant cell death in eye progenitors [47]. The role of the *dpp* pathway in preventing progenitor cell death we have discovered would, in principle, connect *ey* function to cell death. At present, we do not have a mechanistic explanation for why eye progenitor cells are prone to die (a tendency akin to the "death by default" state of cells proposed by Raff [48]), or how *dpp* signaling prevents it, but these are very relevant questions worth pursuing. However, organ size control does not come for free: It requires extra energy expenditure in generating progenitors that are going to undergo apoptosis. Note that these progenitors are perfectly normal cells: If their death is prevented genetically, they become incorporated into eyes (that is why blocking apoptosis results in larger eyes). Our model also suggests that, in order to reproduce the experimental data, the rate of apoptosis should be the noisiest variable. Therefore, precise control of eye size requires an apparently wasteful and noisy mechanism.

Is this control mechanism, based on regulating the rate of cell death of proliferating progenitors, working beyond the eyes? Although not tested formally, the result from Neto-Silva and colleagues, showing that wing asymmetry increased dramatically in mutants for the apoptosis regulator *hid* [49], suggests that apoptosis regulation may be involved in the control of developmental stability of wings too. Beyond *Drosophila*, developmental apoptosis is very frequent [50] and, in some cases, associated to the regulation of cell number, with the mammalian immune cells [51] and the nervous system [52] being striking examples. Therefore, it seems very possible to us that mechanisms similar to the one we have described here will use the feedback modulation of apoptosis as a means not only to controlling organ size but also size robustness in other developmental systems as well.

## Materials and methods

### Method details

***Drosophila* strains and genetic manipulations.** The *optix2/3-GAL4* driver (abbreviated "*optix>*") is expressed in undifferentiated eye progenitors [53]. Other strains used were *UAS-*

*tkvRNAi* (VDRC #102319), *UAS-DadRNAi* (VDRC #108340), *UAS-Dad* [29], *UAS-RHGRNAi* [22], and *UAS-Upd* [54]. The *optix2/3-GAL4*, *UAS-tkvRNAi/CyO* recombinant strain was generated by standard genetic methods. Other strains used were the *dpp* transcriptional reporter *dpp-Z* [55], the *Bar* mutant on the FM7c balancer chromosome (Flybase: FBba0000009), the wild-type strain *Oregon-R (Or-R)*, and the eye-specific driver *GMR-GAL4* (FlyBase: FBgn0020433). The *UAS-mCherry* RNAi line (Bloomington stock centre, reference BDSC_35785) was used as a neutral UAS-RNAi line. The *GMR-Upd* line is described in [38].

**Immunofluorescence and confocal imaging.** Fixation and immunofluorescence of eye primordia was done as in [56]. Imaging was carried out on a Leica SPE confocal setup. Images were then processed using ImageJ. Primary antibodies used were rabbit *anti*-phospho-histone H3 (pH3) used at 1/1,000 (Sigma), rabbit against the cleaved (active) form of Dcp 1 (Dcp-1) used at 1/200 (ASP216, Cell Signaling Technology), and rabbit anti-cleaved (active) caspase 3 (Cas3*) used at 1/500 (D175, Cell Signaling Technology). Fluorescently labeled secondary antibodies were from Molecular Probes and used at 1/1,000. Rhodamine-phalloidin (Life Technologies) was used at 1/400 in some experiments to counterstain the tissue. DAPI (4′,6-diamidino-2-phenylindole, 1/10,000) was used in some experiments to counterstain nuclei. All primary and secondary antibodies were diluted in PBT (PBS + 0.1% Triton X-100).

## Quantification and statistical analysis

**Quantification of adult eye size.** Heads of 1- to 3-day old adult males and females were dissected in PBS, cleared with 15% hydrogen peroxide, and mounted in Hoyer's medium: lactic acid (1:1) [57]. Frontal and occipital planes of each head were photographed using a 10× objective on a Leica DM5000B microscope and a Leica DFC490 digital camera. The areas of the eyes and head were measured using the polygonal tool of ImageJ [58] and expressed as arbitrary units. The supporting figure to Materials and methods (S1 Materials) illustrates the areas as measured. The area of each eye was obtained as the sum of the area of the eye in the frontal and occipital planes.

**Quantification of mitotic and apoptotic rates.** To quantify the apoptosis (detected with anti-Dcp-1) and mitosis (detected with anti-PH3) rates as the Dcp-1/PH3 signal relative to the area anterior to the morphogenetic furrow (that is, differentiation wave front), we wrote a plug-in for ImageJ (included as S1 Macros). Briefly, it works as follows: First, for each sample, the images of a 3D confocal stack are projected along the Z-axis using the maximum pixel intensity. Then, a region of interest (ROI) is drawn by hand to define the region of the eye disc anterior to the morphogenetic furrow, which contains the undifferentiated eye cells. A threshold for pixel intensity is chosen manually on the green channel (Dcp-1/PH3), making sure that the thresholded area includes accurately all the signal. This threshold level is kept for every sample of the series, which were acquired with exactly the same microscope configuration. Finally, the thresholded area is projected onto the original image for the mean intensity value and integrated density to be measured.

## Statistical analysis

The statistical analyses were supported on *R* software [59], and hypothesis decisions were made under 95% confidence threshold in all cases.

## Modeling for intergroups comparison

The metrics of interest were modeled depending on genotype group and sex factors by fitting a median regression model (quantile regression: *rq* function from *quantreg* library for factors) where normality and homoscedasticity assumptions are not required [60]. This model allows

estimating the effect of the different covariables and factors in the median of the response variable, thus allowing the prediction of the median response variable for each group as a sum of the corresponding effects [61]. In our context, the *optix>+* group was set as reference level, and genotype group was coded as multiple dummy factors of presence/absence of each genotype. Thus, the effect of simultaneous down-regulation of 2 genes is also evaluated as an interaction term in the model. Models considering or not sex as a factor were compared by calculating F-statistic of the ratio of variances explained by each fitted model (*anova* function). Sex effect turned out to be nonsignificant regardless of the response variable evaluated; thus, the general model is denoted by the following expression:

$$y_i = \beta_{0;0.5}(optix) + \beta_{1;0.5}\,RHGRI_i + \beta_{2;0.5}\,tkvRI_i + \beta_{3;0.5}\,tkvRI_iDadRI_i + \beta_{4;0.5}\,DadRI_i$$
$$+ \beta_{5;0.5}\,tkvRI_iRHGRI_i + \beta_{6;0.5}\,Bar_i + \beta_{7;0.5}\,Bar_iRHGRI_i + \beta_{8;0.5}Bar_i\,DadRI_i + u_{i,0.5}$$

where

$y_i$ is the i-th value of rE or %Apop variables;

$\beta_{j,0.5}$ is the j-th value of the estimated conditioned coefficient to the 0.5 quantile; and

$u_{i,0.5}$ is the i-th value of the error variable conditioned to the 0.5 quantile.

## Comparison of sFAi variances of each genotype versus control

The sFAi is an intraindividual measure of eye size precision. It is calculated as the relative differences between intraindividual (left and right) eye areas:

$$sFA_i = (L - R)/(L + R)$$

We compared intergroup sFAi through their variances since sFAi distributions are theoretically centered on 0. First, we assume there are no intragroup variance differences due to sex (Levene test median centered; *LeveneTest function* from *car* package). Second, median was subtracted for each group–sex combination to perform a sex-independent comparison. Finally, Levene test median centered was performed for pairwise comparisons of each group against the control one (*optix>+)*. P.values were adjusted by Benjamini–Hochberg method (*LeveneTest* function from *car* package and *p.adj* function, *method = "BH")*.

## Comparison of rE variances of each genotype versus control

rE variance is an interindividual measure of size precision in eye development. First, we assumed there were no intragroup variance differences due to sex (Levene test median centered; *LeveneTest function* from *car* package). Second, median was subtracted for each group–sex combination to perform a sex-independent comparison. Finally, Levene test median centered was performed for pairwise comparisons of each group against the control one (*optix>+)*. P.values were adjusted by Benjamini–Hochberg method (*LeveneTest* function from *car* package and *p.adj* function, *method = "BH")*.

## Supporting information

**S1 Fig. Expression of the apoptosis markers activated caspase-3 (Cas3\*) and activated Dcp-1 in control eye discs.** (**a**, **a'**) *optix>GFP* primordium stained for GFP, the photoreceptor marker Elav and Cas3\*. The expression of the *optix-GAL4* driver, as detected by GFP-expression is outlined (in **a'**). The Cas3\* signal is low (**a'**). (**b**, **b'**) *optix>+* primordium stained for the G2-marker Cyclin B (CycB) and Dcp-1. Dcp-1 signal is detected in a band of cells anterior to the differentiating wavefront (this latter lacks CycB expression).
(PDF)

**S2 Fig. The average ommatidial size is approximately constant in different genotypes tested.** The average ommatidial size was calculated as the area occupied by 10 adjacent ommatidia in the equatorial region of the eye divided by 10. Independent eyes were measured on photographs of mounted eyes of the indicated genotypes. The statistical comparison of the values obtained relative to the control *("optix>+")* did not detect significant differences among genotypes. The data underlying the graphs shown in the figure can be found in "S2_Fig 1_data" in the Supporting information file S1 Raw Data.
(PDF)

**S3 Fig. Study of the effect of a neutral RNAi on rE and sFAi values.** Results are shown for rE and sFAi of eyes from individuals of the following genotypes: "+": Oregon-R (Or-R) wild-type strain; "*optix>+*": Progeny from the cross *optix-GAL4* to Or-R; "*optix>UAS-CherryRI*": The *optix-GAL4; UAS-Cherry_RNAi* progeny obtained by crossing *optix-GAL4* to *UAS-Cherry_R-NAi*. This dataset was obtained independently of the datasets included in Fig 2 of the main text. The median rE of *optix>CherryRI* eyes is slightly higher than that of the *optix>+* reference control group; however, it is not significantly higher than that of the wild-type *Or-R* group (**a**). There are no significant differences in sFAi between any of the groups compared (**b**). The table shows the statistical analysis of rE and sFAi among the different groups considered. This analysis concludes that there is no significant effect of driving a nonspecific, neutral RNAi on eye size precision and that the effect on final eye size is within the variability present among the alternative control lines used (An equivalent table is explained in S2 Table. Further details on the statistical analysis can be found in the main text Materials and methods section). The data underlying the graphs shown in the figure can be found in "S3_Fig 1_data" in the Supporting information file S1 Raw Data.
(PDF)

**S4 Fig. Phenotypic rescue of genotypes with 2 UAS constructs cannot be explained by GAL4 "dilution effect".** We compared the distributions and median values of rE (**a**) and sFAi (**b**) from *optix>tkvRI* (1 UAS transgene: UAS-*tkvRI*) and *optix>tkvRI* + GFP (2 UAS transgenes: UAS-*tkvRI and* UAS-GFP). In the case of eye size (**a**), if the presence of 2 UAS sequences had titrated the GAL4 molecules (effectively halving the number of GAL4 molecules per UAS sequence), the expectation would be a weakening of the phenotype (derived from a weaker expression of *tkv-RNAi*). In the case of sFAi (**b**), titration of GAL4 would result in a reduction of the asymmetry. However, these expectations were not observed. We used the Bayes Factor (BF; shown in plot) as a measure of the strength with which a null hypothesis is accepted or rejected. Values close to 0 allow accepting the null hypothesis: "There is no GAL-4 dilution effect." Black bars show the minimum detectable differences in each comparison depending on sample sizes and distribution shapes with 80% power (see S1 Statistical Methods). Indeed, we find very low BF values for both rE and sFAi, supporting the idea that there is no dilution effect of GAL4 when 2 UAS transgenes, instead of one, are present in the genotype. Therefore, the phenotypic effects detected are genuinely caused by the genetic perturbation. The data used in the graphs shown in the figure can be found in "S4_Fig 1_data" in the Supporting information file S1 Raw Data.
(PDF)

**S5 Fig. Increased cell death and fragmented *dpp* expression in *Bar* discs.** L3 *dpp-Z* discs (**a**, **b**), stained for apoptosis (dcp-1, green), *dpp* transcription (beta-galactosidase, red), and the retinal differentiation marker Elav (blue). (**a**) Control ("+": Oregon-R) and (**b**) *Bar* mutant discs. The ellipse approximately outlines the eye region. "*a*" marks the antennal primordium.
(PDF)

**S6 Fig. Analysis of intraindividual eye size correlation.** The intraindividual FA (FAi) is calculated using the left and right eyes *from the same individual*. The interindividual FA (FAn) is calculated as the difference between the left eye of an individual and the right eye randomly chosen from the population of measured eyes. The difference between FAn and FAi was computed for each fly (ΔFA). If there were no "individual" effect, ΔFA should be 0. The effect of each genotype on ΔFA median, their p.values, and the predicted median for each of them are shown (see S1 Statistical Methods). The ΔFA median is slightly, though significantly greater than 0 across all genotypes relative to the control (*optix>+*). This result indicates that eye size within an individual is slightly correlated irrespective of genotype. The data used in the graphs shown in the figure can be found in "Fig 1_S6Fig 1_data" in the Supporting information file S1 Raw Data.
(PDF)

**S7 Fig. Attenuation of the Dpp signaling pathway does not affect the mitotic rate of eye progenitors.** (**a**, **b**) Representative late L3 eye imaginal discs, stained for the mitotic marker phosphor-histone H3 (PH3), and counterstained with rhodamine phalloidin ("actin"). (**c**) Distribution of mitotic rate, measured as the rate of PH3-positive area relative to the % of available area (nonapoptotic) in the anterior domain (regions outlined in yellow) in the 2 genotypes (see S1 Statistical Methods). Bayes Factor (BF) value close to 0 allows accepting the null hypothesis: "There is no significant differences between genotypes." BF = 0.37. The bar indicates an 80% power to detect differences. This analysis indicates that mitotic rate of progenitors does not change in *optix>tkvRI* relative to *optix>+* controls (see S1 Statistical Methods). The data used in the graphs shown in the figure can be found in "S1_Fig 2_data_A" and "S1_Fig 2_data_B" in the Supporting information file S1 Raw Data.
(PDF)

**S8 Fig. Caspase 3-associated apoptosis is induced by Dad-mediated Dpp signal attenuation.** *optix2/3-GAL4; UAS-Dad ("optix>Dad")* eye disc stained for activated caspase 3 ("cas3\*"). Abundant cas3\* signal is detected in cells anterior to the differentiation wave-front (dashed line). The disc is counterstained with the nuclear marker DAPI. (**b**, **b'**) Close-up of a cas3\*-positive region. Cas3\* signal (**b'**: arrows) overlaps with pycnotic nuclei stained with DAPI (small, dense DAPI signal), indicating that cells enter an irreversible apoptotic process.
(PDF)

**S9 Fig. A mutant condition in which eye size varies dramatically preserving precision.** (**a**, **b**) Heads from GMR>+ (**a**) and GMR>Upd (**b**) adult males; frontal view. While the median eye size (rE: **c**) of *GMR>Upd* flies is approximately 1.4 times that of control (*GMR>+*) ones (statistically significant), the fluctuating asymmetry index ("precision") (FAi: **d**) is indistinguishable between the 2 genotypes. Female (F) and male (M) distributions are shown. (**e**, **f**) Eye primordia stained for the apoptotic marker Dcp-1. Apoptotic signal is observed in both *GMR>Upd* as well as in *GMR>+* control primordia. The data used in the graphs shown in the figure are found in "S1_Fig 4_data" in the Supporting information file S1 Raw Data.
(PDF)

**S1 Table. Analysis of the effect of genotype on rE median, rE dispersion, and sFAi dispersion.** First block shows each genotype's effect on the rE median, their p.values, and the predicted median for each of them. The second and third blocks show the standard deviation of each group for rE and sFAi, respectively, and the result of the comparison of variances of each group against the *optix>+* reference group.
(PDF)

**S2 Table. Analysis of genotype effect on % of apoptotic area in the anterior domain of the eye primordium (%Apop).** Each genotype effect on %Apop median, their p.values, and the predicted median for each of them are shown (see main text, Materials and methods section). (PDF)

**S1 Materials.** Eye and head areas as measured in this work. (**a**, **a'**) Frontal (**a**) and occipital (**a'**) focal planes, with the areas corresponding to the left eye outlined using the "polygon" tool of ImageJ. The area of the eye is the sum of both areas. (**b**) Area of the head, outlined using the "lasso" tool. The area of each eye was obtained as the sum of the area of the eye in the frontal and occipital planes (a + a'). (PDF)

**S1 Statistical Methods. This file describes in detail the statistical methods used for data analysis.** (PDF)

**S1 Mathematical Model. Full mathematical model including its deterministic and stocastic versions, the analysis of effects of noise on the different variables of the model, and a sensitivity analysis.** (PDF)

**S1 Macros. Macro 1. Measure green channel.** Macro for immunostaining quantification, designed to measure the proportion of signal of a "green-labeled" marker (in this paper PH3 or Dcp-1) to a selected are (region of interest) from confocal imaging data. Can be run with ImageJ. Measures the green signal. It runs on all maximal projection images, as.tif files, within a folder. **Macro 2. Measure green channel_threshold individual.** Macro for immunostaining quantification, designed to measure the proportion of signal of a "green-labeled" marker (in this paper, PH3 or Dcp-1) to a selected are (region of interest) from confocal imaging data. Can be run with ImageJ. As Macro 1 Measure green channel, but asks for a threshold signal for each image. **Macro 3. Measure green channel_imagen individual.** Macro for immunostaining quantification, designed to measure the proportion of signal of a "green-labeled" marker (in this paper, PH3 or Dcp-1) to a selected are (region of interest) from confocal imaging data. Can be run with ImageJ. As S1 Macro Measure green channel, but for a single image. **Macro 4. Genera Z projections.** Macro for immunostaining quantification, designed to measure the proportion of signal of a "green-labeled" marker (in this paper, PH3 or Dcp-1) to a selected are (region of interest) from confocal imaging data. Can be run with ImageJ. Helps with bulk processing: If the data is a.lif within a folder, it allows selection of the folder and then extracts all files archives from the.lif and generates a maximal projection of each of them. (ZIP)

**S1 Raw Data.** The file "S1_Raw_Data.zip" is a compressed folder that contains individual tables including the raw data (quantifications) for each figure (main and supporting) plus a.txt file named "Figures_rawdata_INFO" in which the columns in each table are described. (ZIP)

## Acknowledgments

We thank Mustafa Khammash, Maike Kittelman, and Alistair McGregor for discussions; Erika Bach for the *GMR-Upd* strain, and ALMIA/CABD (especially Jose M. Urbano), for imaging support.

## Author Contributions

**Conceptualization:** Saúl Ares, Fernando Casares.

**Formal analysis:** Tomas Navarro, Javier Muñoz-García, Saúl Ares.

**Funding acquisition:** Saúl Ares, Fernando Casares.

**Investigation:** Tomas Navarro, Antonella Iannini, Marta Neto, Paulo S. Pereira, Saúl Ares, Fernando Casares.

**Methodology:** Tomas Navarro.

**Project administration:** Fernando Casares.

**Software:** Alejandro Campoy-Lopez.

**Supervision:** Fernando Casares.

**Visualization:** Tomas Navarro, Antonella Iannini, Marta Neto, Saúl Ares, Fernando Casares.

**Writing – original draft:** Saúl Ares, Fernando Casares.

**Writing – review & editing:** Tomas Navarro, Marta Neto, Paulo S. Pereira, Saúl Ares, Fernando Casares.

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
