## [Editor Report · Decision Letter 0]

6 Apr 2023

Dear Fernando, 

Thank you for submitting your manuscript entitled "Feedback control of organ size precision is mediated by BMP2-regulated apoptosis in the Drosophila eye" for consideration as a Research Article by PLOS Biology.

Your manuscript has now been evaluated by the PLOS Biology editorial staff as well as by an academic editor with relevant expertise and I am writing to let you know that we would like to send your submission out for external peer review. However, we have decided to consider it as a Short Report, as we think it will fit better in that format (https://journals.plos.org/plosbiology/s/what-we-publish#loc-short-reports). Please note that in this format manuscript can only have up to 4 figures, thus you will have to convert the rest into supplementary figures and select that type of article when you submit the metadata (see below).

Before we can send your manuscript to reviewers, we need you to complete your submission by providing the metadata that is required for full assessment. To this end, please login to Editorial Manager where you will find the paper in the 'Submissions Needing Revisions' folder on your homepage. Please click 'Revise Submission' from the Action Links and complete all additional questions in the submission questionnaire.

Once your full submission is complete, your paper will undergo a series of checks in preparation for peer review. After your manuscript has passed the checks it will be sent out for review. To provide the metadata for your submission, please Login to Editorial Manager (https://www.editorialmanager.com/pbiology) within two working days, i.e. by Apr 10 2023 11:59PM.

Kind regards,

Ines

--

Ines Alvarez-Garcia, PhD

Senior Editor

PLOS Biology

---

## [Decision Letter · Decision Letter 1]

7 Jun 2023

Dear Fernando,

Thank you for your patience while your manuscript entitled "Feedback control of organ size precision is mediated by BMP2-regulated apoptosis in the Drosophila eye" was peer-reviewed at PLOS Biology as a Short Report. Please also accept again my apologies for the delay in providing you with our decision. The manuscript has now been evaluated by the PLOS Biology editors, an Academic Editor with relevant expertise, and by two independent reviewers. 

The reviews are attached below. As you will see, the reviewers find the conclusions potentially interesting, but they also raise several concerns that would need to be addressed to strengthen the results. Reviewer 1 suggests several experiments to confirm the role of Dpp in apoptosis regulation, and Reviewer 2 points out several potential flaws in the modelling, mainly regarding the choice of parameters and the type of simulations performed, among other issues.

After discussing the reviews with the Academic Editor, we have decided to invite you to revise the work to thoroughly address the reviewers' reports. Given the extent of revision needed, we cannot make a decision about publication until we have seen the revised manuscript and your response to the reviewers' comments. Your revised manuscript is likely to be sent for further evaluation by all or a subset of the reviewers.

**IMPORTANT - SUBMITTING YOUR REVISION**

3. Resubmission Checklist

a) *PLOS Data Policy*

b) *Published Peer Review*

Sincerely,

Ines

--

Ines Alvarez-Garcia, PhD

Senior Editor

PLOS Biology

Reviewers' comments

Rev. 1:

The manuscript by Casares, Ares and colleagues uses the Drosophila eye as model system and combines genetic manipulations and mathematical modelling to identify a role of Dpp in controlling eye size by blocking apoptosis in progenitor cells, to propose a role of apoptosis in reducing size variability and to generate a theoretical model to show how this newly identified mechanism contributes to reducing such variability. I am an experimentalist that appreciates mathematical modeling of biological processes but without sufficient background to review this part of the manuscript in a thorough manner. I have several issues with respect to the experimental part and their conclusions related to experiments and modeling that should be addressed to demonstrate and reinforce their proposal. I have only some minor comments to reinforce the role of Dpp in controlling eye size by reducing apoptosis (see at the end of the list below). However, I have major issues with the implication of Dpp and apoptosis in size precision. The two major comments listed below are fundamental to demonstrate their proposal.

Major (essential) comments:

(1) Authors claim that in all their experimental conditions (eg. different genetic manipulations to block apoptosis or reduce or activate Dpp signaling) size precision is lost as a result of increased variability and from these observations they propose that apoptosis plays a major role in the process. Is not the reduction in size precision a simple consequence of the developmental system being far away from the wild type (most probably, the most stable and precise) condition? Have the authors analyzed other unrelated mutant conditions where there is a clear effect on eye size and/or patterning but without affecting size precision? I think this should be addressed by the authors to avoid any trivial explanation to their results and model.

(2) Have authors produced progenitors in excess and checked whether apoptosis is increased, whether this apoptosis can be rescued by increasing Dpp signaling and whether this has a (negative) impact on size precision? This is genetically feasible and funfdamental to test their hypothesis.

Other (essential) comments:

(1) I wonder why authors have not tried to deplete Dpp and see whether the effects on apoptosis are reproduced. This is important for the model as different BMPs and/or sources (can) exist.

(2) Apoptosis: authors should check with other tools whether cells are actually dying upon Dpp signaling inhibition (eg. TUNEL, pyknosis)

(3) Cell size should be considered in their quantifications. As a proxy, the number of ommatidia could be quantified to monitor whether the reduction in eye size correlates with a proportional reduction in ommatidia number

Minor comments

(1) Is RHGRi a polycistronic RNAi or a synthetic miRNA?

(2) Apoptosis index should be defined in the text.

(3) Dcp1 antibody is against the cleaved (active) form of Dcp1? Details are lacking in the text and M&M

(4) P values should be shown in the graphs (and not only as a supplementary table)

Rev. 2:

The manuscript addresses the problem of final size control of developing tissues, particularly robustness to developmental noise, and uses the Drosophila eye imaginal disc as a model system. The question is very appropriate and much of the work is very well done.

The paper has two parts, an experimental section followed by a modeling section. The experimental section provides strong evidence that Dpp produced by cells in the optogenetic furrow inhibits the death of progenitor cells. Noting that genetic manipulations that affect this circuit lead to greater variation in eye size (both between left and right and between individuals), they postulate that Dpp's effect is part of a control circuit that serves to control eye growth and differentiation in a manner that buffers against developmental noise. It's a reasonable hypothesis, although enthusiasm should perhaps be tempered a bit by the fact that the appearance of excessive variation in response to mutations of all kinds is so common a phenomenon in biology that the now-famous term "decanalization" was coined by Waddington to describe it. Still, this reviewer has no objection to giving this hypothesis a good test, and applauds the authors for seeking to do so.

The rest of the paper describes the development and analysis of an explicit model to support the hypothesis. This is an appropriate use of modeling, and a good deal of sophisticated analysis is carried out. However, there are a number of flaws and problems with the modeling approach that likely impact the conclusions. In particular, one gets the sense that many of the assumptions made to get the model to work and enable the application of analytical tools were sufficiently ad hoc as to call the conclusions into question. Troubling assumptions include the notion that one can replace signaling over space with saturating functions of cell numbers and that developmental noise can be appropriately characterized using Langevin equations.

Specific comments on the modeling section include:

1. The diagram (Figure 2 of the supplementary section) and discussion do not match the differential equations. In the text and figure they state that "Rn cells maturate automatically into R cells." However, in their equations, the rate of Rn cell maturation into R cells is a function of R cells; in fact, it is the same function of R cells that is meant to represent Hh's effects on G cell differentiation into Rn cells. Whether this dependence was intentional or not is unclear, however given the parameter ranges explored it certainly has a large effect on the behavior of the equations, and therefore needs to be justified. In particular, there would need to be a strong justification for using the identical dependence of Rn differentiation on R as is used for the dependence of G differentiation on R. While that might seem reasonable at first, the geometry of the system ensures that the average distance of G cells from R cells is very different from the average distance of Rn cells from R cells; if a dependence on R is supposed to capture the notion that Hh only diffuses a certain distance, then one would expect to see very different dependences.

2. The authors use a non-spatial (ODE) model to represent a phenomenon that plays out in space, and therefore must incorporate a variety of simplifying assumptions that are quite ad hoc, namely three different Hill functions with arbitrary Hill coefficients and EC50 values. The desire to use ODEs to approximate what is really a PDE problem is understandable, and is occasionally justifiable, but a lot more work is required to justify this. This is because the authors seek not only to produce a model that agrees in its broad qualitative behavior with observations, they also wish to examine things like robustness and sensitivity to noise. In the end, it might be easier to at actually write out the PDEs, put in diffusion explicitly, and do a few spatial simulations, than to try to make a convincing argument that one can substitute Hill functions with numbers of cell instead.

3. Parameter choices are arbitrary and may be unrealistic in some cases. For example, the size of the R domain over which Hh effects reach half-saturation is set at 190µm. This reviewer is not aware whether the range of Hh spread has been measured in the eye disc, but elsewhere it tends to have a very short range, typically on the order of a few cell diameters. How would changing K_r to, say, 4 µm change the behavior of the model?

4. Modeling "developmental noise" is not a straightforward process. In proliferating cell systems, the primary source of developmental noise would seem to be the fact that one has finite numbers of cells making discrete decisions probabilistically; thus the same initial conditions will lead to different outcomes when simulated multiple times. Monte Carlo simulations provide the most direct way to enumerate outcomes, but they can be a bit cumbersome to run, and difficult to analyze. Langevin noise approximations provide a faster way to explore noise effects, and enable the use of analytical tools, but they are not always appropriate for every purpose. They are often used together with chemical-type equations composed of linear terms; with proliferation equations (where species are growing exponentially) they can underestimate noise effects. In view of this it would have been a good idea to supplement the data with a few well-chosen Monte Carlo simulations.

5. A biologically useful control system should not only reject noise disturbances, it should also exhibit an appropriate degree of parametric robustness, i.e. it should not require that parameters—including initial conditions—be fine-tuned in order to produce a reliable output. Whereas the authors may feel that a full exploration of the robustness properties of their model is beyond the scope of this paper, it behoves them to show that the model they come up with is at least not excessively fragile.

---

## [Decision Letter · Decision Letter 2]

14 Nov 2023

Dear Fernando,

Thank you for your patience while we considered your revised manuscript entitled "Feedback control of organ size precision is mediated by BMP2-regulated apoptosis in the Drosophila eye" for consideration as a Short Report at PLOS Biology. Please accept again my apologies for the long delay in sending you our decision. Your revised study has now been evaluated by the PLOS Biology editors, the Academic Editor and the two original reviewers. 

The reviews are attached below. As you will see, the reviewers appreciate all the work that has been done during the revision, however they also raise several remaining issues that would need to be addressed. Reviewer 1 is mostly satisfied, raising one point that you will need to clarify. Reviewer 2 thinks that while the model shows that basic qualitative behaviours to determine size precision can be reproduced, it doesn’t explain why the model has the impact on size precision that it has or why all the pieces of the model are necessary to achieve size precision as it’s claimed.

After discussing the comments with the Academic Editor, we would like to invite you to submit a revision that addresses the remaining points raised by the reviewers. To address Reviewer 2’s concerns, the revision should include a section in the discussion outlining the assumptions you make in the model, the reasons why these assumptions are made, and stating clearly that the model is limited because there are still a lot of unknowns in this complex biological system. You could also consider adding a table to illustrate the assumptions if it helps to clarify, and comment on the approximation of spatial aspects of eye development used in the model, as pointed out by Reviewer 2.

We expect to receive your revised manuscript within 1 month. Please email us (plosbiology@plos.org) if you have any questions or concerns, or would like to request an extension. We will assess your revised manuscript and your response to the reviewers' comments with our Academic Editor aiming to avoid further rounds of peer-review. Please also make sure to address the data and other policy-related requests stated below.

*Published Peer Review History*

*Press*

Sincerely,

Ines

--

Ines Alvarez-Garcia, PhD

Senior Editor

PLOS Biology

DATA POLICY:

Many thanks for including the data underlying the graphs shown in the figures. Please indicate in each corresponding figure legend (both main and supplementary) where the data can be found. For example, you can add at the end of the figure legends: "The data underlying the graphs shown in the figure can be found in file X."

Please also make sure they are referred to (in the manuscript, figure legends, and the Description field when uploading your files) using the following format verbatim: S1 Data, S2 Data, etc. The excel files should be saved using exactly the following convention: S1_Data.xlsx (using an underscore).

CODE POLICY

Please ensure that the code is sufficiently well documented and reusable, and that your Data Statement in the Editorial Manager submission system accurately describes where your code can be found.

Reviewers' comments:

Rev. 1:

I went through the responses and the manuscript and I think authors have addressed successfully my concerns.

However, I have still one question (an important one, I think). How do the authors explain the rescue of FAi by co-depletion of Tkv and Dad? In this case, the sensor element (Dpp receptor) is lost but Dpp pathway activity rescued. Some discussion to address this question would suffice as an important element is lost in the feedback model that senses variability in size and induction of apoptosis.

Rev. 2:

The authors have attempted to either resolve or rebut the criticisms of the previous review. I do appreciate the argument that sometimes, in modeling, one has to make simplifying assumptions and insert phenomenological terms in order to make one's model tractable for analysis. However, it must be noted that such practice is primarily justified when the purpose of the modeling is to capture broad, qualitative behaviors. To the extent that the authors wish to represent that their model generates the basic behaviors of a system that grows, has a differentiation wave, and then stops, and qualitatively reproduces mutant phenotypes, there isn't too much problem: their model does these things. But to be fair, the main reason it does these things seems to be because they've put pieces into their equations to ensure that it does. These are typical phenomenological terms—elements inserted into the equations to produce a desired behavior (thresholds, saturation, steep response etc.), rather than terms that correspond to known physiology. There are quite a lot of these, more than one often sees in models of comparable complexity. Of course, one can explore model behavior in asymptotic regimes (i.e. really high or really low levels of variables) to convince an audience that the details of these phenomenological pieces are qualitatively unimportant, and indeed these sorts of explanations are to be found in several places in the manuscript (although mostly buried in supplemental information and not particularly well organized—the letter of response to reviews is clearer in this regard than the manuscript itself). 

The problem is that the authors don't portray the modeling as merely showing that basic qualitative behaviors of the system can be reproduced. They argue that the model explains not only "average" mutant phenotypes, but also the degree of individual phenotypic variation across experimental conditions. Even though the model reproduces that behavior, there is no way to determine whether that is because the generic structure of this model guarantees it, or whether it is entirely dependent on the details of the phenomenological pieces they've put in (which, by definition, don't correspond in any simple way to actual mechanisms). For example, when I requested previously that the authors consider using a spatial framework in their equations, it was not because I just wanted them to do extra work, but because I am concerned that the ability of the model to capture the effects of different perturbations on size precision might depend on the way in which they used arbitrary threshold functions as a rough substitute for the effects of space. 

In the end, despite having strong experimental data, is just too much of a stretch to argue that the model they derive constitutes an "explanation" for the size precision that is observed experimentally in growing eye discs.

---

## [Editor Report · Decision Letter 3]

27 Nov 2023

Dear Fernando,

Thank you for the submission of your revised Short Report entitled "Feedback control of organ size precision is mediated by BMP2-regulated apoptosis in the Drosophila eye" for publication in PLOS Biology. On behalf of my colleagues and the Academic Editor, Nic Tapon, I am delighted to let you know that we can in principle accept your manuscript for publication, provided you address any remaining formatting and reporting issues. These will be detailed in an email you should receive within 2-3 business days from our colleagues in the journal operations team; no action is required from you until then. Please note that we will not be able to formally accept your manuscript and schedule it for publication until you have completed any requested changes.

PRESS

Sincerely, 

Ines

--

Ines Alvarez-Garcia, PhD

Senior Editor

PLOS Biology
